# Experiments on Liquid Flow through Non-Circular Micro-Orifices

**DOI:** 10.3390/mi11050510

**Published:** 2020-05-19

**Authors:** Stefano Cassineri, Andrea Cioncolini, Liam Smith, Michele Curioni, Fabio Scenini

**Affiliations:** 1Materials Performance Centre, Department of Materials, University of Manchester, Oxford Road, Manchester M13 9PL, UK; stefano.cassineri@manchester.ac.uk (S.C.); liam.smith-6@postgrad.manchester.ac.uk (L.S.); michele.curioni@manchester.ac.uk (M.C.); 2Department of Mechanical, Aerospace and Civil Engineering, University of Manchester, George Begg Building, Sackville Street, Manchester M1 3BB, UK

**Keywords:** micro-orifice, micro-fluidics, non-circular, square, rectangular, experiment, turbulent flow, discharge, micro-electro-mechanical system, MEMS

## Abstract

Microfluidics is an active research area in modern fluid mechanics, with several applications in science and engineering. Despite their importance in microfluidic systems, micro-orifices with non-circular cross-sections have not been extensively investigated. In this study, micro-orifice discharge with single-phase liquid flow was experimentally investigated for seven square and rectangular cross-section micro-orifices with a hydraulic diameter in the range of 326–510 µm. The discharge measurements were carried out in pressurized water (12 MPa) at ambient temperature (298 K) and high temperature (503 K). During the tests, the Reynolds number varied between 5883 and 212,030, significantly extending the range in which data are currently available in the literature on non-circular micro-orifices. The results indicate that the cross-sectional shape of the micro-orifice has little, if any, effect on the hydrodynamic behavior. Thus, existing methods for the prediction of turbulent flow behavior in circular micro-orifices can be used to predict the flow behavior in non-circular micro-orifices, provided that the flow geometry of the non-circular micro-orifice is described using a hydraulic diameter.

## 1. Introduction

Microfluidics is the branch of fluid mechanics that deals with fluid flow in constrained channels of sub-millimeter dimensions. Due to the recent progress in micro-fabrication and manufacturing techniques, microfluidics is now an active research area with practical applications in mechanical engineering, chemistry, biology, and medicine. For example, microfluidics underpins the design of microelectromechanical systems (MEMS) [1,2] and, in particular, the design of micro-cooling systems [3,4,5,6], microvalves and micropumps [7,8] and micromixers [9,10]. Microfluidics provides the foundation for lab-on-chip microdevices, which integrate chemical or biological laboratories onto a single chip, and of organ-on-chip technologies, where human cells are cultured in microfluidics devices to simulate the physiology of entire organs. Some of these approaches hold the promise of revolutionizing drug discovery and development [11,12], early disease diagnosis [13], and biochemical analysis [14,15,16].

Micro-orifices, in particular, are extensively used in microfluidics systems such as micro-evaporators and micro-heat sinks, microinjectors, micropumps and microvalves. For example, micro-orifices are used to suppress flow instabilities in micro-evaporators [17] and to control the flow in vapor compression refrigeration systems [18,19]. Micro-orifices are also used in corrosion studies [20,21,22,23], in flow metering [24] and for organic matter synthesis [25]. Due to its practical relevance, orifice microfluidics has been investigated extensively, both for single-phase and two-phase flow.

The focus of this work is on the single-phase flow of liquids through micro-orifices, which is characterized by the Reynolds number Re and the dimensionless pressure drop K:(1)$$Re=\frac{\rho \;V\;d_{hyd}}{\mu }$$
(2)$$K=\frac{2\;\Delta P}{\rho \;V^{2}}$$
where ρ and μ are the liquid density and viscosity, V is the average flow velocity through the micro-orifice, ΔP is the pressure drop across the micro-orifice, and $d_{hyd}$ is the micro-orifice hydraulic diameter (four times the flow area divided by the wetted perimeter, i.e., $d_{hyd}=4A_{flow}/P_{wet}$). Therefore, if the orifice cross-section is circular, then $d_{hyd}=d$, where d is the circular orifice diameter. A summary of the available experimental studies on single-phase liquid flow through micro-orifices is provided in Table 1.

The studies by Mishra and Peles [29], Tu et al. [31], and Cioncolini et al. [36], which are focused on cavitating two-phase flow in micro-orifices, are included in Table 1 because they also provide experimental data on single-phase liquid flow. As it is evident from Table 1, data are available for Reynolds number values in the range between 0.1 and 220,000, i.e., from creeping flow to fully turbulent flow conditions. Water, refrigerants and oil have been used as test fluids with micro-orifices, mostly with circular cross-sections; square cross-section micro-orifices have been investigated to a lesser extent. During testing, the micro-orifices are normally installed in tubes of flow area much larger than that of the orifice. This is the case for the studies summarized in Table 1, as can be noted from the small values of the orifice to tube diameter ratio (below about d/D≈0.2). Consequently, the influence of the orifice to tube diameter ratio on the flow through the micro-orifice is normally negligible or absent. In most cases, the available data have been obtained for thick micro-orifices (i.e., with an orifice thickness to diameter ratio t/d>0.5); this is different from the normal practice for studies on macro-orifices (where, normally, t/d<0.5). As a result, micro-orifices are sometimes referred to as short-tube micro-orifices. In microfluidics systems, thick orifices are much more commonly used because thin orifice plates might suffer from excessive mechanical deformation under the differential pressure load. This might be one of the fundamental reasons for the limitations of macroscale orifice prediction methods when extrapolated to micro-systems. Thus, methods specifically intended for micro-orifices are needed.

Available prediction methods for micro-orifice flow include the empirical correlation of Cioncolini et al. [33,34] for turbulent flow conditions and the analytical method developed by Dagan et al. [37] for creeping flow conditions; see Equations (3) and (4), respectively.
(3)$$K=3.137\;Re^{-0.0737}for\;1000<Re<220,000\;and\;\frac{t}{d}<7$$
(4)$$K=\frac{12\pi }{Re}\;\left(1+\frac{16}{3\pi }\;\frac{t}{d}\right)for\;Re\to 0^{+}\;and\;0<\frac{t}{d}<2$$

As can be noted from the above equations, in turbulent flow conditions the dimensionless pressure drop only depends on the Reynolds number, whereas with creeping flows it also depends on the orifice thickness to diameter ratio. The experimental database that underpins Equation (3) includes 266 data points from five literature studies [27,28,31,33,34], which are fitted by Equation (3) with a mean absolute percentage error of 8.9%. As discussed by Szolcek et al. [35], Equation (4) is valid for Reynolds numbers below about Re≈100−200 and for thickness to diameter ratios up to t/d=27.

In most cases, the micro-orifices manufactured by microfabrication techniques for use in MEMS technology have non-circular cross-sections. However, as it is evident from Table 1, the majority of the investigations to date have been carried out using circular cross-section micro-orifices. Only Wang et al. [28] and Mishra and Peles [29] investigated non-circular micro-orifices with square cross-sections, and only covering the relatively limited range of Reynolds number values between 160 and 4500. Therefore, the main objective of this work is to provide new data on the flow through micro-orifices with non-circular cross-sections, focusing in particular on turbulent flow conditions at high Reynolds numbers where, at present, there are no data. Specifically, we tested seven non-circular micro-orifices with square or rectangular cross-sections, with hydraulic diameters ranging between 326 µm and 510 µm. Using ambient temperature (298 K) and high temperature (503 K) pressurized water (12 MPa) as the test fluid, we explored Reynolds numbers in the range of 5883–212,030 that significantly extend the range covered to date in turbulent flow conditions.

In practical applications, the flow through non-circular micro-orifices is normally predicted by extrapolating from methods originally designed for circular micro-orifices, such as Equations (3) and (4). Practically, this is achieved by substituting the hydraulic diameter $d_{hyd}$ in place of the diameter d. Unfortunately, there is no data to support or validate such an extrapolation beyond the square cross-section micro-orifice investigations summarized in Table 1. An additional objective of this work, therefore, is to assess the suitability of the turbulent flow prediction method in Equation (3) for predicting square and rectangular cross-section micro-orifice data, for use in practical design applications.

The rest of this paper is organized as follows: the micro-orifice manufacturing, the test setup and experimental methods are presented in Section 2, while the results are presented and discussed in Section 3.

## 2. Materials and Methods

The geometric details of the eight micro-orifices (one circular and seven non-circular) used here are summarized in Table 2, while scanning electron microscope (SEM) images of all the disks containing the micro-orifices are provided in Figure 1.

Each micro-orifice is manufactured from a 12 mm diameter stainless steel disc of 1.1 mm nominal thickness. Three micro-orifices (Samples 2, 3, and 4) have square cross-sections and four micro-orifices (Samples 5, 6, 7, and 8) have rectangular cross-sections, with aspect ratios varying between one and 2.26. One additional micro-orifice (Sample 1) was manufactured with a circular cross-section to validate the experimental setup. Sample 1, with a circular cross-section, was manufactured by conventional mechanical drilling. Conversely, the other samples were manufactured via spark erosion using a copper die that was manufactured via Electrical Discharge Machining (EDM). Specifically, a copper rod 12 mm in diameter was first electrically ablated via a +GF+ AgieCharmilles FI 440 ccS wire cut EDM machine to obtain a copper electrode with the same cross-section of the wanted micro-orifice. Then, via +GF+ AgieCharmilles form 20 sinker EDM machining, the stainless-steel discs were spark eroded to create the non-circular holes. A dual step process involving a rough cut and then a precision finish was required to obtain the desired micro-orifice shape. Finally, the front and back of each sample were ground up to 600 grit using Si-C emery paper.

Roughness effects, which can be important with distributed pressure drops in tubes and channels, are normally of minor importance in orifices (and other localized discontinuities) where the pressure loss is concentrated, not distributed, and mostly affected by the contraction and expansion of the flow. Moreover, measuring the surface roughness inside micro-orifices as small as those used here does not seem feasible. Conversely, while not directly measured, the surface roughness of the front and back faces of the samples can be estimated to be in the order of 1 µm based on the material used to manufacture the samples (stainless steel 304) and the P600 Silicon Carbide grinding paper used for the surface finishing [38,39]. It is worth noting that the SEM images in Figure 1 were taken before grinding the samples: after grinding, the samples were inspected but no SEM images were taken.

As can be noted in Figure 1, the non-circular micro-orifices have rounded corners, and have therefore rounded square or rectangular cross-sections. Whilst this is of no concern in measuring the aspect ratio, it is evident that the flow area and wetted perimeter must be measured by taking the rounded corners into account. In practical terms, this was achieved by converting the SEM images in Figure 1 into binary images and then applying standard image-processing techniques using the free software GNU Octave [40]. From the calibration of the SEM microscope, the error in measuring the linear dimensions of the micro-orifices was estimated to be in the order of ≈10 µm. For simplicity, the errors on the wetted perimeter, flow area, and hydraulic diameter included in Table 2 were estimated by neglecting the rounded corners and assuming that the micro-orifice cross-section was rectangular (see Appendix A for details). The absolute errors in Table 2 correspond to a relative error in the wetted perimeter within 2–4%, a relative error in the flow area within 4–7%, and a relative error in the hydraulic diameter within 5–8%.

Schematic representations of the flow cell test section and of the flow loop are provided in Figure 2. The flow cell test section consisted of the 12 mm diameter disc containing the micro-orifice, which was placed on the top of a copper gasket and mounted between a Swagelok (Solon, OH, USA) male tube adapter (3/8 in. tube OD × 3/8 in. male ISO parallel thread) and a Swagelok female connector (1/4 in. Tube OD × 3/8 in. female ISO parallel gauge thread). The sealing between the micro-orifice disc and the Swagelok female connector was achieved by compressing the male tube adapter onto the water flow entry side of the micro-orifice disc. The flow cell test section was connected to a recirculating flow loop comprising a 13-litre autoclave and a 250-litre feed-tank, capable of delivering micro-filtered and ultra-high purity water (conductivity of 0.005 µS/cm) at a variable pressure (0.1-20.0 MPa) and variable temperature (from ambient temperature up to 633 K). The test water contained 2 ppm of Li in solution to inhibit the corrosion product deposition frequently occurring in flow restrictions with high-temperature water flow [23,41]. A high-pressure diaphragm pump was used to provide the driving head for the flow, and a pulsation damper was used to ensure pulsation-free flow conditions during all the experiments. On the return line of the autoclave, an in-line mixed-bed Lithium exchange resin was used to remove any contamination arising from the natural corrosion of the recirculating flow loop.

The pressure drop across the micro-orifices was measured using a differential pressure transducer (by Rosemount Inc., Shakopee, MN, USA; model C2051CD, ±2.07 MPa span, 0.075% full-scale accuracy) for pressure drop values up to about 2.0 MPa in magnitude. For pressure drops in excess of 2.0 MPa, two absolute pressure transducers (by Omega Engineering Inc., Norwalk, CT, USA model PXM 409, 0–24.5 MPa span, 0.1% full-scale accuracy) were used to measure the static pressures upstream and downstream of the micro-orifices, and the final pressure drop was calculated as the difference between the upstream and the downstream static pressure values. Prior to the tests, all pressure transducers were calibrated offline at a certified metrological laboratory. Overall, the experimental uncertainty of the pressure drop was in the range of 0.1–10%. The mass flow rate was measured using two variable-area digital flow meters (by Swagelok, model VAF-M2, 0–1.4 cm^3^/s and 1.1–11.1 cm^3^/s spans) that were calibrated end-to-end online prior to the tests (accuracy within ±1%). The water temperature inside the autoclave was measured (to within ±1 K) using direct immersion type-J thermocouples (not shown in Figure 2).

During testing, the operating pressure of the flow loop was maintained at (12.0 ± 0.1) MPa, and measurements were carried out at an ambient temperature (298 ± 1) K and high temperature (503 ± 5) K. Clearly, the measurements at high temperature were carried out to widen the Reynolds number range covered, by exploiting the reduction in viscosity with the increasing operating temperature typical of water and most liquids. The density and viscosity of the test water were calculated with NIST-REFPROP [42] and their values are provided in Table 3.

The Reynolds number and the dimensionless pressure drop were calculated as indicated in Equations (1) and (2). The experimental uncertainties, estimated via standard single-sample error propagation [43], are in the range of 2–4% for the Reynolds number and 8–14% for the dimensionless pressure drop. During the experiments, the Mach number of the flow through the micro-orifices (ratio of the average flow velocity through the micro-orifice to the sound velocity in the test water) never exceeded 0.1, so compressibility effects can be ignored. The test rig was validated with the circular cross-section micro-orifice (Sample 1), as discussed in the following section.

## 3. Results and Discussion

The raw data are presented in Figure 3, where the measured pressure drop across the micro-orifices is plotted as a function of the mass flow rate (error bars would be of similar size as the markers, and are therefore not included). Notably, the data show the approximately quadratic dependence of the pressure drop on the mass flow rate, which is typical of turbulent flow in pipes or channels. It is evident that, for each micro-orifice tested, the pressure drop measured at high temperature is higher than that measured at ambient temperature for the same mass flow rate. This is a consequence of the reduction in water density with increasing temperature. At a high temperature, due to the reduction in density, a given mass flow rate corresponds to a higher flow velocity, and therefore a higher pressure drop.

The quadratic relationship between pressure drop and mass flow rate evident in Figure 3 indicates the absence of any cavitation occurring during the test; in fact, it is well known that, when cavitation occurs, the discharge deviates substantially from the quadratic relationship characteristic of single-phase turbulent flow. In particular, when cavitation occurs, the flow chokes, the downstream pressure no longer affects the flow and, correspondingly, the mass flow rate becomes proportional to the square root of the upstream pressure [36]. The dimensionless parameter normally used to characterize cavitating flows through micro-orifices is the cavitation number σ, defined as follows:(5)$$\sigma =\frac{2\;\left(P_{down}-P_{v}\right)}{\rho V^{2}}$$
where $P_{down}$ is the static pressure downstream of the micro-orifice, whilst $P_{v}$ is the local vapor pressure. Cavitation inception depends on the size and geometry of the micro-orifice, and it is normally observed for cavitation numbers in the range of 0.1–1 [29,36,44,45]. In the present tests, the cavitation number ranged between 1.02 and 141 (corresponding to average flow velocities through the micro-orifice of 13–92 m/s)—these values were large enough to confidently exclude the occurrence of cavitation.

The measurements are presented in dimensionless form in Figure 4, where the dimensionless pressure drop is plotted versus the Reynolds number. The micro-orifice prediction method for turbulent flow conditions in Equation (3), extrapolated using the hydraulic diameter in the Reynolds number as indicated in Equation (1), is also included in Figure 4. The trend of the dimensionless pressure drop for the circular cross-section micro-orifice (Sample 1) decreases slightly with the increasing Reynolds number. As can be noted, the measurements compare favorably with the reference prediction method: all data points do not depart by more than one to two error bars from Equation (3). This validates the present experimental apparatus and procedure. Regarding the non-circular micro-orifice data in Figure 4, the slightly decreasing trend of the dimensionless pressure drop with increasing Reynolds number is less evident than for the circular cross-section in Sample 1, suggesting that the shape of the cross-section may have an effect on the flow through the micro-orifice. Such an effect, however, appears to be rather small in magnitude. In fact, all non-circular micro-orifice data points do not deviate by more than one to two error bars from the reference prediction method in Equation (3).

The present data are plotted in Figure 5a, together with data from the literature from Table 1 [27,28,31,33,34,36]. Overall, the scatter in the present non-circular data appears to be comparable to the scatter in the data from the literature. This indicates that any effect of the micro-orifice cross-sectional shape on the flow does not significantly alter the non-circular micro-orifice discharge when compared with the discharge of a circular micro-orifice at the same Reynolds number. The measured dimensionless pressure drop values are compared with the predictions of Equation (3) in the parity plot provided in Figure 5b. Notably, almost all measurements are within ± 30% from the corresponding predictions. Overall, Equation (3) fits the present non-circular micro-orifice data with a mean absolute percentage error of 10.8%, which is comparable to the present experimental error.

Within the limits of the present study, the turbulent flow through non-circular micro-orifices does not differ significantly from the turbulent flow through circular micro-orifices. Cross-sectional shape effects, if at all present, appear to be within the experimental errors, and therefore are of minor concern in practical applications. Existing prediction methods such as Equation (3) can be used in practical applications, providing the geometry of the non-circular micro-orifice is described using the hydraulic diameter.

## 4. Conclusions

We measured the discharge behavior of seven square and rectangular cross-section micro-orifices with hydraulic diameters between 326 and 510 µm, using pressurized water (12 MPa) at ambient temperature (298 K) and high temperature (503 K). In our tests, we investigated Reynolds numbers between 5883 and 212,030, significantly extending the range of Reynolds numbers for which data are available in the case of non-circular micro-orifices. Within the limits of this study, we concluded that the flow is not significantly affected by the cross-sectional shape of the micro-orifice. The existing discharge prediction method proposed by Cioncolini et al. [33,34] for turbulent flow in circular micro-orifices can be used to predict the flow with non-circular micro-orifices, provided the flow geometry is described using the hydraulic diameter.

All the data discussed here are directly presented in the paper in graphical form and therefore are automatically accessible.

## Figures and Tables

**Figure 1 micromachines-11-00510-f001:**
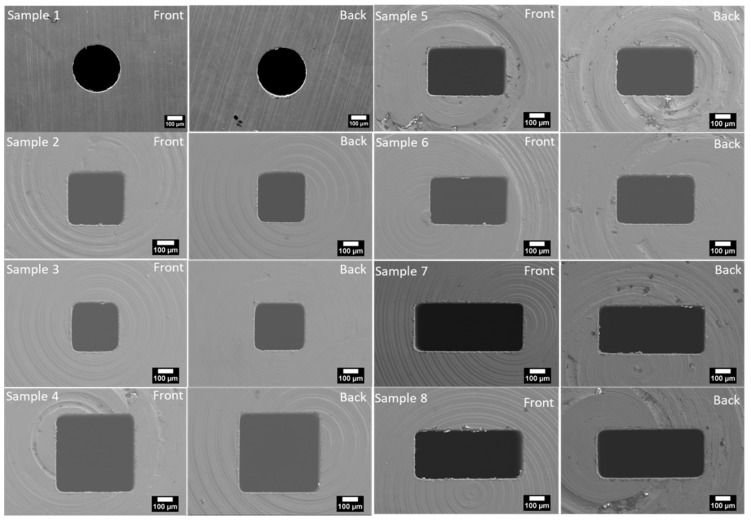
SEM images (front and back) of all the discs containing the micro-orifice.

**Figure 2 micromachines-11-00510-f002:**
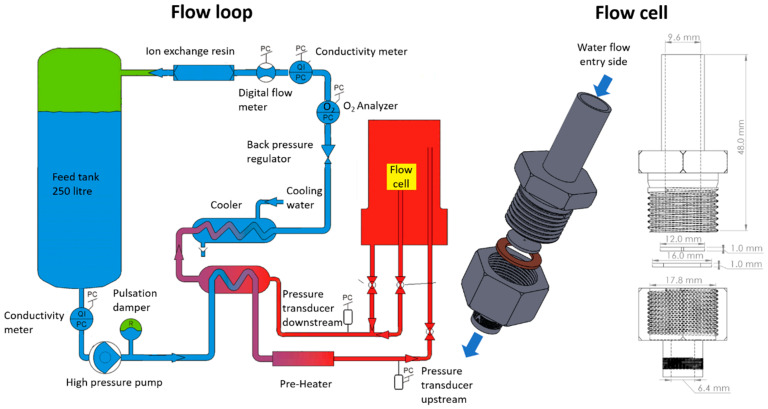
Schematic representation of the flow loop (**left**) and 3D representation and drawing of the flow cell test section (**right**).

**Figure 3 micromachines-11-00510-f003:**
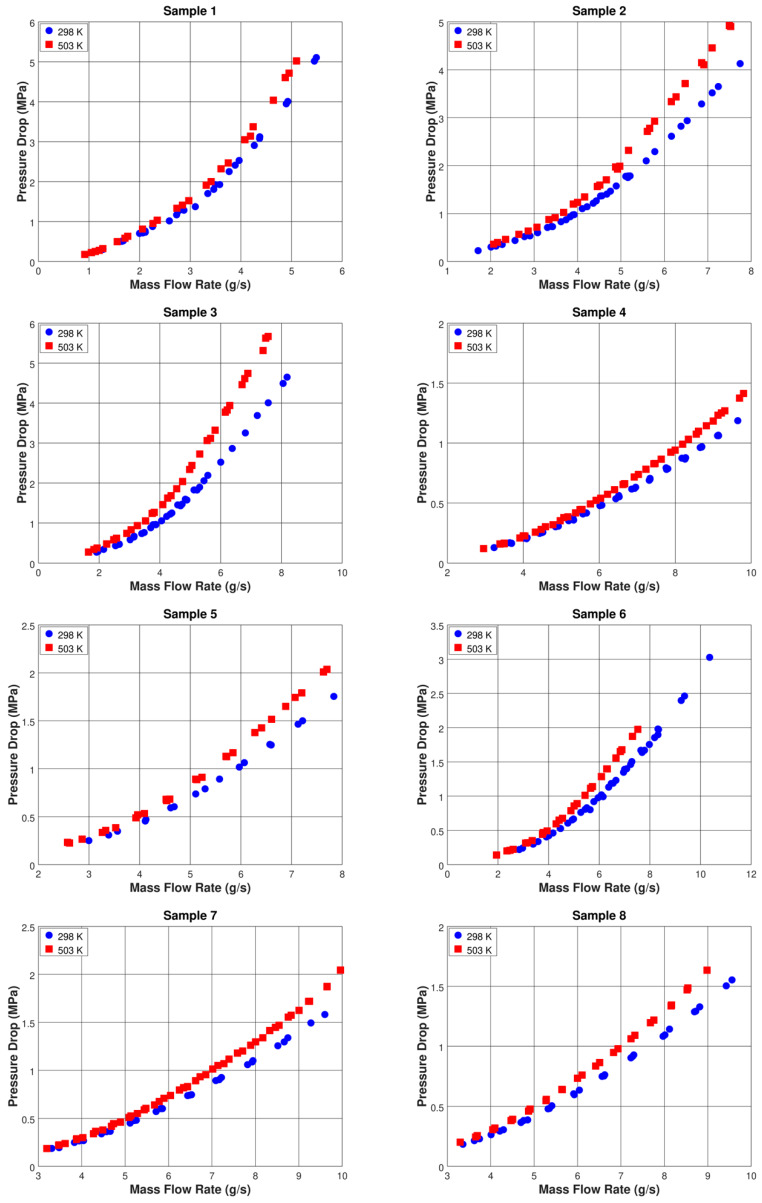
Discharge characteristics of the micro-orifices tested.

**Figure 4 micromachines-11-00510-f004:**
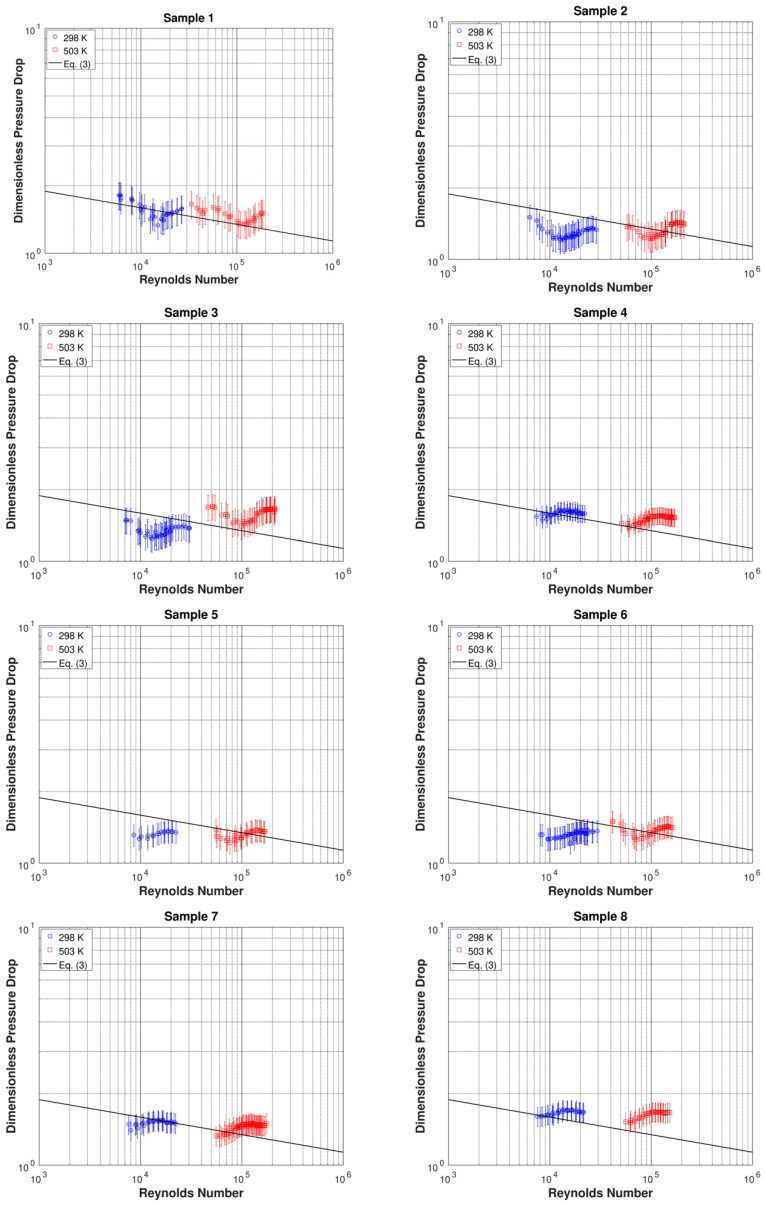
Dimensionless pressure drop as a function of Reynolds number for the micro-orifices tested.

**Figure 5 micromachines-11-00510-f005:**
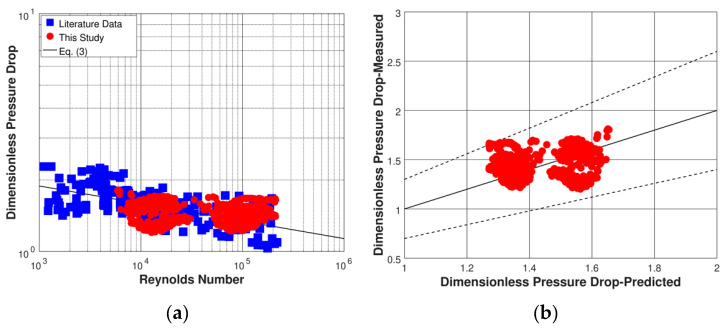
(**a**) Dimensionless pressure drop as a function of Reynolds number for the micro-orifices tested in this study and from the literature; (**b**) parity plot of measured dimensionless pressure drop vs. prediction of Equation (3) (the dashed lines are ±30% bounds).

**Table 1 micromachines-11-00510-t001:** Experimental data bank for micro-orifice liquid flow.

Reference	$d_{hyd}\;(\mu m)$	$d_{hyd}/D$	$t/d_{hyd}$	Fluid	Reynolds	Cross-Section
Johansen [26]	704; 1634	0.09; 0.209	0.083	Oil	0.1–150	Circular
Kojasoy et al. [27]	1000; 2000	0.057; 0.114	1.0; 2.0	R113	560–14,000	Circular
Wang et al. [28]	150; 370	NA	NA	Water	800–4500	Square
Mishra and Peles [29]	11.5	0.114	1.7	Water	160–550	Square
Phares et al. [30]	81.7–59.2	0.008–0.016	2.65–5.16	Water	2.5–120	Circular
Tu et al. [31]	31.0; 52.0	0.007; 0.012	2.5; 4.2	R134a	1600–6500	Circular
Ushida et al. [32]	100; 400	NA	0.05; 0.2	Water	1.3–1300	Circular
Cioncolini et al. [33]	150–600	0.015–0.06	1.87–6.93	Water	6000–26,000	Circular
Cioncolini et al. [34]	300; 600	0.0306; 0.0612	1.67; 3.33	Water	18,000–220,000	Circular
Szolcek et al. [35]	200	0.02	4.25–27.0	Water	5–4500	Circular
Cioncolini et al. [36]	150; 300	0.015–0.03	3.53–6.93	Water	3425–30,043	Circular
This study	326–510	0.05–0.08	2.29–3.62	Water	5883–212,030	Square Rectangular

**Table 2 micromachines-11-00510-t002:** Dimensions of the micro-orifice samples.

Sample No.	$P_{wet}\;(\mu m)$	$A_{flow}\cdot {\mathrm{10}}^{-3}\;(\mu m^{2})$	$d_{hyd}\;(\mu m)$	t (mm)	$t/d_{hyd}$	$AR{\;}^{a}$	Cross-Section
1	927 ± 40	68.3 ± 4.6	295 ± 24	1.05 ± 0.01	3.56	NA	Circular
2	1210 ± 40	98.6 ± 6.0	326 ± 23	1.18 ± 0.01	3.62	1.06	Square
3	1209 ± 40	99.7 ± 6.0	330 ± 23	1.19 ± 0.01	3.61	1.05	Square
4	1958 ± 40	249.6 ± 9.8	510 ± 23	1.17 ± 0.01	2.29	1.00	Square
5	1571 ± 40	153.7 ± 7.8	391 ± 22	1.16 ± 0.01	2.97	1.58	Rectangular
6	1586 ± 40	155.6 ± 7.9	392 ± 22	1.24 ± 0.01	3.16	1.58	Rectangular
7	1946 ± 40	209.1 ± 9.7	430 ± 22	1.06 ± 0.01	2.46	2.23	Rectangular
8	1993 ± 40	221.9 ± 9.9	445 ± 22	1.11 ± 0.01	2.49	2.26	Rectangular

^a^ Aspect ratio of the micro-orifice cross-section: ratio of the long side length to the short side length.

**Table 3 micromachines-11-00510-t003:** Density and viscosity of the water at 298 and 503 K.

Property	298 K	503 K
ρ (kg/m^3^)	1002	835.9
μ (µPa s)	891.6	118.7

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
