# Peer review of "Experiments on Liquid Flow through Non-Circular Micro-Orifices"

_micromachines, 2020, doi:10.3390/mi11050510_

Round 1
Reviewer 1 Report
The authors investigated fluid flow in micro orifices. They presented experimental data obtained from 8 different orifices. They made careful experiments and experimental data seem to be reliable. There are certain issues/comments to be considered to improve the technical content:
1-While at first glance there seem to exist few studies on fluid flow on micro orifices, there are some cavitation studies on micro orifices, which include experimental results corresponding to flow conditions prior to cavitation inception. These studies will be useful for the authors. They should be included in Literature review and the authors can benefit from the experimental data as well as discussion about the results. This will improve the technical content.
2-Roughness effects should be elaborated in more detail in this study. Since the diameter is varied in this study, it will be nice to sort the devices according to relative roughness and use this parameter in discussing the results.
3-There are a number of correlations recommended for friction factors in short channels (with developing flow effects). They are very useful. It will be worthwhile to compare the results with the predictions of these correlations.
4-The authors should also elaborate on the transition from laminar to turbulent flows. Size and roughness effects should be discussed in detail.
5-The authors operated at high pressures. At those pressure, cavitation could also incept and affect the results. The authors should provide more clarification.
6-A better and detailed image displaying flow path of the flow along microorifices (with significant dimensions) would provide a better idea to the readers.
Author Response
1-While at first glance there seem to exist few studies on fluid flow on micro orifices, there are some cavitation studies on micro orifices, which include experimental results corresponding to flow conditions prior to cavitation inception. These studies will be useful for the authors. They should be included in Literature review and the authors can benefit from the experimental data as well as discussion about the results. This will improve the technical content.
As a matter of fact, two studies (Mishra and Peles [29] and Tu el al. [31]) addressing cavitation that also include single-phase liquid flow data in usable form are already included in the literature survey and experimental databank in Table 1. Following the reviewer’s suggestion, one more study has been included in the revised manuscript and the data added to the databank in Table 1 (the plots have been correspondingly updated to reflect this). The following comment has been included in the revised manuscript to clarify why these studies are included:
The studies by Mishra and Peles [29], Tu et al. [31], and Cioncolini et al. [36], which are focused on cavitating two-phase flow in micro-orifices, are included in Table 1 because they also provide experimental data on single-phase liquid flow.
2-Roughness effects should be elaborated in more detail in this study. Since the diameter is varied in this study, it will be nice to sort the devices according to relative roughness and use this parameter in discussing the results.
To address the reviewer’s comment, the following paragraph has been included in the revised manuscript:
Roughness effects, which can be important with distributed pressure drop in tubes and channels, are normally of minor importance with orifices (and other localized discontinuities) where the pressure loss is concentrated, not distributed, and mostly affected by the contraction and expansion of the flow. Moreover, measuring the surface roughness inside micro-orifices as small as those used here does not seem feasible. Conversely, albeit not directly measured, the surface roughness of the front and back faces of the samples can be estimated to be on the order of 1 mm, based on the material used to manufacture the samples (stainless steel 304) and the P600 Silicon Carbide grinding paper used for the surface finishing [38]. It is worth noting that the SEM images in Figure 1 were taken before grinding the samples: after grinding the samples were inspected but no SEM images were taken.
3-There are a number of correlations recommended for friction factors in short channels (with developing flow effects). They are very useful. It will be worthwhile to compare the results with the predictions of these correlations.
The correlation used here (Equation 3) is very simple in structure because it is a power law with just one dimensionless group and two empirical parameters, and already fits the present databank with a mean absolute percentage error comparable with the measuring error in the data. We therefore see limited incentive/usefulness in considering other, more complicated correlations: the correlations for pressure drop with developing flow in short tubes that we are aware of typically combine different contributions to account for the sudden contraction of the flow, the sudden expansion of the flow, and friction in the throat.
4-The authors should also elaborate on the transition from laminar to turbulent flows. Size and roughness effects should be discussed in detail.
The laminar to turbulent flow transition with micro-orifices has not been extensively investigated, so it is not possible to provide any complete account of it at present. This is however outside the scope of the present study, which is concerned with turbulent flow in non-circular micro-orifices, and not with the laminar to turbulent flow transition. As far as we are aware of, there are no studies that investigate the effect of surface roughness, or that systematically investigate size effects on the laminar to turbulent flow transition in micro-orifices.
5-The authors operated at high pressures. At those pressure, cavitation could also incept and affect the results. The authors should provide more clarification.
To address the reviewer’s comment, the following paragraph is now included in the revised manuscript:
The quadratic relationship between pressure drop and mass flow rate evident in Figure 3 indicates the absence of any cavitation occurring during the tests, as it is well known that when cavitation occurs then the discharge deviates substantially from the quadratic relationship characteristic of single-phase turbulent flow. In particular, when cavitation occurs the flow chokes, the downstream pressure no longer affects the flow, and correspondingly the mass flow rate becomes proportional to the square root of the upstream pressure [36]. The dimensionless parameter normally used to characterize cavitating flows through micro-orifices is the cavitation number defined as follows:
[see Equation (5) in the revised manuscript]
where is the static pressure downstream of the micro-orifice, whilst is the local vapor pressure. Cavitation inception depends on the size and geometry of the micro-orifice, and is normally observed for cavitation numbers in the range of 0.1-1 [29,36,42,43]. In the present tests, the cavitation number ranged between 1.02 and 141 (corresponding to average flow velocities through the micro-orifice within 13-92 m/s): values that are large enough to exclude the occurrence of cavitation.
Even though the relationship between pressure drop and mass flow rate was always quadratic, a few data points (20 points out of 543) had cavitation number slightly below 1 (in the range of 0.8-1): in order to stay on the safe side, these points have been excluded from the databank (all plots included in the revised manuscript have been regenerated to account for this). The conclusions in the manuscript are not affected by this modification.
6-A better and detailed image displaying flow path of the flow along microorifices (with significant dimensions) would provide a better idea to the readers.
To address the reviewer’s comment, the micro-orifice test piece schematics in Figure 1 has been replaced with a 3D CAD drawing (with dimensions).
Reviewer 2 Report
- Viscosity and density are essentially important in estimating flow property. The author should show in the paper how to decide them and provide their values at temperatures tested.
- Cavitation may occur in such high velocity flow as examined in the paper. Velocity should be given and discuss the effect of cavitation. Increase in pressure drop at higher temperature may be caused by cavitation.
Author Response
1) Viscosity and density are essentially important in estimating flow property. The author should show in the paper how to decide them and provide their values at temperatures tested.
To address the reviewer’s comment, the following paragraph is now included in the revised manuscript:
Density and viscosity of the test water were calculated with NIST-REFPROP [41] and their values are provided in Table 3:
Table 3. Density and viscosity of the test water at 298 and 503 K
|
|
298 K |
503 K |
|
(kg/m3) |
1002 |
835.9 |
|
(mPa s) |
891.6 |
118.7 |
2) Cavitation may occur in such high velocity flow as examined in the paper. Velocity should be given and discuss the effect of cavitation. Increase in pressure drop at higher temperature may be caused by cavitation.
To address the reviewer’s comment, the following paragraph is now included in the revised manuscript:
The quadratic relationship between pressure drop and mass flow rate evident in Figure 3 indicates the absence of any cavitation occurring during the tests; in fact, it is well known that when cavitation occurs then the discharge deviates substantially from the quadratic relationship characteristic of single-phase turbulent flow. In particular, when cavitation occurs the flow chokes, the downstream pressure no longer affects the flow, and correspondingly the mass flow rate becomes proportional to the square root of the upstream pressure [36]. The dimensionless parameter normally used to characterize cavitating flows through micro-orifices is the cavitation number defined as follows:
[see Equation (5) in the revised manuscript]
where is the static pressure downstream of the micro-orifice, whilst is the local vapor pressure. Cavitation inception depends on the size and geometry of the micro-orifice, and is normally observed for cavitation numbers in the range of 0.1-1 [29,36,42,43]. In the present tests, the cavitation number ranged between 1.02 and 141 (corresponding to average flow velocities through the micro-orifice within 13-92 m/s): these values were large enough to confidently exclude the occurrence of cavitation.
Reviewer 3 Report
This is an interesting paper describing experimental studies of flow through non-circular (rectangular and square) micro-orifices over a high Reynolds (Re) number range. In addition, the authors compare the results with previous studies on circular micro-orifices over a similar Re range. They find that the dimensionless pressure drop behavior is comparable for systems where the hydraulic diameter of the non-circular cross sections is similar to the diameter of the circular cross-sections. The authors also show that the empirical correlation describing the dimensionless pressure drop under turbulent conditions for circular micro-orifices holds for non-circular micro-orifices when described with the hydraulic diameter.
The authors present a very complete analysis with a through explanation and quantification of contributions to experimental error. This paper reads well and can be considered for publication upon consideration of the following minor comments.
- In the introduction on page 2 lines 56-70, the authors describe important dimensions of the micro-orifice that include the tube diameter and the thickness. In addition, the authors point out how typical dimensions of these parameters compared with the orifice have important distinction with typical macro-orifice dimensions. It would be useful to the reader if a schematic was provided to better visualize these dimensions on the orifice design.
- On page 6, line Figure 4 line 208, I believe you are referring to Figure 5 (right), not Figure 4 (right).
Author Response
1) In the introduction on page 2 lines 56-70, the authors describe important dimensions of the micro-orifice that include the tube diameter and the thickness. In addition, the authors point out how typical dimensions of these parameters compared with the orifice have important distinction with typical macro-orifice dimensions. It would be useful to the reader if a schematic was provided to better visualize these dimensions on the orifice design.
To address the reviewer’s comment, the micro-orifice test piece schematics in Figure 1 has been replaced with a 3D CAD drawing (with dimensions).
2) On page 6, line Figure 4 line 208, I believe you are referring to Figure 5 (right), not Figure 4 (right).
Typo corrected
Round 2
Reviewer 1 Report
The authors addressed most of the reviewer's comments. The manuscript is acceptable.
Reviewer 2 Report
This paper is available to application in micro-orifices. The author responds well to my comments, and I agree to publication.